# Dynamic Submodular Maximization

**Morteza Monemizadeh**
Department of Mathematics and Computer Science
TU Eindhoven, the Netherlands
m.monemizadeh@tue.nl

## Abstract

One of the basic primitives in the class of submodular optimization problems is the submodular maximization under a cardinality constraint. Here we are given a ground set $V$ that is endowed with a monotone submodular function $f : 2^V \to \mathbb{R}^+$ and a parameter $0 < k \leq n$ and the goal is to return an optimal set $S \subseteq V$ of at most $k$ elements, i.e., $f(S)$ is maximum among all subsets of $V$ of size at most $k$. This basic primitive has many applications in machine learning as well as combinatorial optimization. Example applications are agglomerative clustering, exemplar-based clustering, categorical feature compression, document and corpus summarization, recommender systems, search result diversification, data subset selection, minimum spanning tree, max flow, global minimum cut, maximum matching, traveling salesman problem, max clique, max cut, set cover and knapsack, among the others. In this paper, we propose the first dynamic algorithm for this problem. Given a stream of inserts and deletes of elements of an underlying ground set $V$, we develop a dynamic algorithm that with high probability, maintains a $(\frac{1}{2} - \epsilon)$-approximation of a cardinality-constrained monotone submodular maximization for any sequence of $z$ updates (inserts and deletes) in time $O(k^2 z \epsilon^{-3} \cdot \log^5 n)$, where $n$ is the maximum size of $V$ at any time. That is, the amortized update time of our algorithm is $O(k^2 \epsilon^{-3} \cdot \log^5 n)$.

## 1 Introduction

A general approach to solve machine learning problems as well as combinatorial optimization problems is to cast the problem at hand as a submodular optimization problem and then solve the submodular problem (approximately) using a rich toolkit of algorithmic techniques known for this class of problems. Such problems include agglomerative clustering, exemplar-based clustering [DF07], categorical feature compression [BCE+19], document summarization [LB11, SSSJ12], recommender systems [EG11], search result diversification [AGHI09], data subset selection [WIB15], social networks analysis [KKT15], minimum spanning tree, global minimum cut, maximum matching, traveling salesman problem, max clique, max cut, set cover and knapsack, among the others.

One of the basic primitives in the class of submodular optimization problems is the submodular maximization under a cardinality constraint. Here we are given a ground set $V$ that is endowed with a monotone submodular function $f : 2^V \to \mathbb{R}^+$ and a parameter $0 < k \leq n$ and the goal is to return an optimal set $S \subseteq V$ of at most $k$ elements, i.e., $f(S)$ is maximum among all subsets of $V$ of size at most $k$. In this paper, we propose the first dynamic algorithm for this problem. We state our main result here:

**Theorem 1** *Suppose we start with an empty set $V$. Then, there exists a randomized dynamic algorithm that with probability at least $1 - \frac{1}{n^2}$ maintains a $(\frac{1}{2} - \epsilon)$-approximation of a cardinality-constrained monotone submodular maximization for any sequence of $z$ updates (inserts and deletes) in $O(k^2 \epsilon^{-3} \cdot \log^5 n)$ amortized update time, where $n$ is the maximum size of $V$ at any time.*

We should mention that classical methods such as the celebrated greedy algorithm [NWF78] or its accelerated versions [BV14] and [MBK$^+$15] require random access to the entire data, make multiple passes, and select elements sequentially in order to produce near optimal solutions. Naturally, such solutions cannot scale to large instances.

Probably the closest work to our work are two recent papers due to Mirzasoleiman, Karbasi and Krause [MKK17] and Kazemi, Zadimoghaddam, and Karbasi [KZK18]. In [MKK17] the authors develop a dynamic streaming algorithm that given a stream of inserts and deletes of elements of an underlying ground set $V$, $(1/2 - \epsilon)$-approximates the submodular maximization under cardinality constraint using $O(d^2(k\epsilon^{-1}\log k)^2)$ space and $O(dk\epsilon^{-1}\log k)$ average update time, where $d$ is an upper-bound for the number of deletes that are allowed. Thus, if the number of deletions is linear in terms of the maximum size of the ground set $V$, that is, at least $\Omega(n)$ deletions, it is in fact better to re-run the known greedy algorithms (say, [NWF78]) after every insertion and deletion.

The follow-up paper [KZK18] studies approximating submodular maximization under cardinality constraint in three models, (1) centralized model, (2) dynamic streaming where we are allowed to insert and delete (up to $d$) elements of an underlying ground set $V$, and (3) distributed (MapReduce) model. In order to develop a generic framework for all the three models, they develop a coreset for the submodular maximization under cardinality constraint. Their coreset has a size of $O(k \log k + d\log^2 k)$. Out of this coreset we can extract a set $S$ of size at most $k$ whose $f(S)$ in expectation is at least $\frac{1}{2}$-approximation of the optimal solution. The time to extract such a set $S$ from the coreset is $O(dk \log^2 k + d\log^3 k)$. Once again, if the number of deletions is $\Omega(n)$, where $n$ is the maximum size of the ground set $V$ at any time $t$, it is in fact better to re-run the known greedy algorithms (say [NWF78]) after every insertion and deletion.

However, our algorithm in Theorem 1 has $\tilde{O}(k^2\epsilon^{-3} \cdot \log^5 n)$ amortized update time which is independent of the number of deletions $d$. Very recently we learned that at NeurIPS 2020 there is another paper titled "Fully Dynamic Algorithm for Constrained Submodular Optimization" due to Lattanzi, Mitrovic, Norouzi-Fard, Tarnawski, and Zadimoghaddam that consider the same problem that we study in this paper. The paper presents a dynamic algorithm whose amortized expected update time is $O(\frac{\log^6(k) \cdot \log^2(n)}{\epsilon^6})$. The amortized expected update time of our algorithm is $O(k^2\epsilon^{-2} \cdot \log^3 n)$. Asymptotically the update time of this algorithm is better than our algorithm. However, in reality these two bounds are incomparable. As an example, for practical values of $k$, say $k \leq 2^{20}$ and for an error bound of $\epsilon \leq 0.05$, the term $\frac{\log^6(k)}{\epsilon^6}$ is approximately $2^{52}$ while the term $k^2\epsilon^{-2}$ in our update time is approximately $2^{45}$ which is smaller than their update time for $n$ even as large as $2^{100}$. Moreover, our algorithm works with high probability and is much simpler than their algorithm. We think we can use our worst case framework in Section 3 to improve their update time from expected to a high probability bound.

Here we mention the main difference between the streaming and the dynamic setting. In the streaming setting the main concern is the space complexity. We often compute a sketch of the input that is revealed in a streaming fashion. At the end of the stream we compute a solution using the sketch that we maintained in the course of stream. On the other hand, in the dynamic setting the main complexity is the time. The idea is that given the input that is revealed in a streaming fashion, we are interested in seeing the solution and the changes in the solution after every insert or delete. The main motivation is for learning highly dynamic and sensitive data (such as time series) that we need to take an action as soon as we see a shift in the function of the underlying data that we observe. Since we need to react to changes in the solution fast, we need to (re)-compute the solution as fast as we can. Indeed, we cannot wait till the end of the stream and take the corresponding action afterwards. The underlying assumption for the dynamic setting is that nowadays with existing machines that can easily have (SD)RAMs of GBs and soon TBs, the space constraint will not be a problem, but the time complexity is the main bottleneck. The results from [MKK17, KZK18] are streaming algorithms whose time complexities depend on the number of deletions (Theorem 1 of the second reference) which will be high if we want to (re)-compute a solution after each insertion or deletion.

**Overview of Proof of Theorem 1.**   An interesting property of a submodular function $f : 2^V \to \mathbb{R}^+$ is that it satisfies $f(A \cup \{e\}) - f(A) \geq f(A \cup \{e\}) - f(A)$ for all $A \subseteq B \subseteq V$ and $e \notin B$. Our main idea is to combine this property with a logarithmic rate of sampling and then greedily collect the heavy items (whose marginal gain are above a threshold) and remove light items (whose marginal

gain are below a threshold) at each rate. Here we let $\Delta_f(e|A) \doteq f(A \cup \{e\}) - f(A)$ be the marginal gain of adding an element $e \in V$ to $A$ where $A \subseteq V$ and $e \in V$.

Let us first consider the offline scenario. We later show how to handle insertion and deletion of elements. Suppose we are given a ground set $V$ of size $n$ endowed with a monotone submodular function $f : 2^V \to \mathbb{R}^+$ under a cardinality constraint parameterized by $0 < k \leq n$. Let $OPT = \max_{S \subseteq V : |S| \leq k} f(S)$ and let $R_0 = V$ and $G_0 = \emptyset$. Let $\tau = \frac{OPT}{2k}$.

We sample a set $S_i \subseteq R_{i-1}$ of $s = O(\epsilon^{-2} \log n)$ elements uniformly at random. That is, we sample each element of $R_{i-1}$ with probability $p = \frac{s}{|R_{i-1}|}$. We let $G_i = G_{i-1}$. We then greedily find those elements of $S_i$ whose marginal gain with respect to the set $G_i$ is at least the threshold $\tau$ and add them to $G_i$. Next, we filter those elements of $R_{i-1} \backslash S_i$ whose marginal gain with respect to the set $G_i$ is below $\tau$ and let $R_i$ be the rest of elements (i.e, those whose marginal gain with respect to the set $G_i$ is at least $\tau$). We then recurse if $R_i$ is not empty. Here the main idea is at each step $i$ of this recursive sampling algorithm, with high probability, we either have $|G_i| \geq |G_{i-1}| + 1$ or $|R_i| \leq \frac{|R_{i-1}|}{2}$. Thus, after $i^* = O(k \log n)$ recursive sampling steps we either have $|G_{i^*}| \geq k$ or we come up with an empty set $R_{i^*}$. We let $G$ be the final set of elements whose marginal gain are above the threshold $\tau$.

Next consider a dynamic scenario where elements are inserted to $V$ or deleted from $V$. Once a new element $e$ is inserted. We loop through steps of the recursive sampling and at each Step $i$, we sample $e$ with probability $p = \frac{s}{|R_{i-1}|}$. If $e$ is sampled, we re-iterate all steps of the recursive sampling from Step $i$ going down to Step $i^*$. Each step $i$ of the recursive sampling consists of one greedy and filtering subroutines and it can be done in $O(|R_{i-1}|k)$. Therefore, since $i^* = O(k \log n)$, the expected computation that is associated to an insertion will be $O(\epsilon^{-2} k^2 \log^3 n)$. The same is true when an element $e \in V$ is deleted. We loop through steps of the recursive sampling and at each Step $i$, we check if $e \in S_i$ which happens with probability $p = \frac{s}{|R_{i-1}|}$. If $e \in S_i$, we re-iterate all steps of the recursive sampling from Step $i$ going down to Step $i^*$. Again, we can show that the expected computation that is associated to a deletion will be $O(\epsilon^{-2} k^2 \log^3 n)$.

To have the dynamic algorithm that works with high probability we create $O(\log n)$ instances of this recursive sampling and run all of them in parallel. After any sequence of $z$ insertions and deletions, we drop those instances whose computations are more than $cz\epsilon^{-2} k^2 \log^3 n$ for some constant $c$. We show that with high probability it remains at least one instance whose total computation is at most $cz\epsilon^{-2} k^2 \log^3 n$. That is, the amortized update time of that instance is $c\epsilon^{-2} k^2 \log^3 n$.

Finally we should mention that for the threshold $\tau$ we choose $\frac{OPT}{2k}$ assuming we know $OPT$. In reality we do not know $OPT$. We can consider two scenarios. The first scenario is when we are given a bound on the maximum element of $V$, that is, say $\max_{e \in V} f(e) = \Theta(\Gamma)$. This is actually a fair assumption that we often make when we generalize the insertion-only streaming model to dynamic streaming models. For example, Frahling and Sohler in [FS05] show that we can find coresets of small size for many clustering problems (a subset of submodular optimization problems) in dynamic geometric streams if we have an upper-bound on the maximum cost of the optimal clustering, something which is not possible if we do not have such an upper-bound. Since $OPT \leq ck \cdot \Gamma$ for a reasonably large constant $c$, we run our recursive sampling algorithm for $\ell \in [0..\epsilon^{-1} \cdot \log(ck\Gamma)]$ guesses $(1 + \epsilon)^\ell$ of $OPT$ and report the best solution of all guesses. This blows up the update time by a factor $\epsilon^{-1} \log(k\Gamma)$ and the approximation factor would be $(1/2 - \epsilon)$.

If we are not given such a bound, we can keep a max heap of the elements that are inserted but not deleted at any time $t$. The insertion and deletion times of the max heap are logarithmic in terms of the number $n$ of items that are stored in the max heap. Finding the maximum $r$ elements stored in the heap can be done in $O(r \log n)$ time. We then do as follows. We sample a set $S$ of $O(\sqrt{nk})$ elements and let $\Gamma = \max_{e \in V} f(e)$ and run the algorithm as for the first scenario. In parallel, at any time $t$, we extract the set $T$ of maximum $O(\sqrt{nk})$ elements from the max heap and run a simple greedy algorithm for $T$. At the end, we report the best solution of these two parallel runs. In this way, the update time increases to $\tilde{O}(\sqrt{nk})$ and the approximation factor would be $(1/2 - \epsilon)$. We elaborate on this second method in the supplementary material.

## 1.1 Preliminaries

Suppose we have a ground set $V$. A function $f : 2^V \to \mathbb{R}^+$ is called *submodular* if it satisfies $f(A \cup \{e\}) - f(A) \geq f(B \cup \{e\}) - f(B)$ for all $A \subseteq B \subseteq V$ and $e \notin B$. When $f$ satisfies the additional property $f(A \cup \{e\}) - f(A) \geq 0$ for all $A$ and $e \notin A$, we say $f$ is *monotone*. We let $\Delta_f(e|A) \doteq f(A \cup \{e\}) - f(A)$ be the marginal gain of adding an element $e \in V$ to $A$ where $A \subseteq V$ and $e \in V$. The term $\Delta_f(e|A)$ is *a discrete derivative* that quantifies the increase in utility obtained when adding $e$ to a set $A$. Observe that, the submodularity condition can be written as $\Delta_f(e|A) \geq \Delta_f(e|B)$ for all $A \subseteq B$.

Monotone submodular maximization under a cardinality constraint for a monotone function $f$ is defined as $OPT = \max_{S \subseteq V : |S| \leq k} f(S)$. We denote by $\mathcal{O}$ an optimal subset of size at most $k$ that achieves the optimal value $OPT = f(S^*)$. Note that we can have more than one optimal set.

The seminal result by Nemhauser, Wolsey and Fisher [NWF78] shows that a simple greedy algorithm can approximate a cardinality constrained monotone submodular maximization problem to a factor of $(1 - 1/e)$ of optimal. This greedy algorithm is as follows. In the beginning, we let $S = \emptyset$. We then take $k$ passes over the set $V$ and in each pass we find an element $e \in V$ that maximizes $\Delta_f(e|S)$, add it to $S$ and delete it from $V$.

## 1.2 Basic primitives

In this paper we frequently use two basic primitives. The first one is a simple greedy algorithm parameterized by a threshold $\tau$ and a set size $k$. Given two sets $S$ and $G$ of elements, the greedy algorithm scrolls through the set $S$ and adds those elements whose marginal gain with respect to the set $G$ is at least $\tau$ provided that the size of $G$ is less than $k$.

The second one is a simple filtering algorithm parameterized by a threshold $\tau$ and a set size $k$. Given two sets $R$ and $G$ of elements, we iterate through the elements of the set $R$ and drop those elements whose marginal gain with respect to the set $G$ is less than $k$.

---

**Basic Primitives**

**Greedy:**
**Input:** Sets $S$ and $G$ and parameters $\tau, k$.
 1: **for** each $e \in S$ **do**
 2:   **if** $\Delta_f(e|G) \geq \tau$ and $|G| < k$ **then**
 3:     $G = G \cup \{e\}$.
**Output:** Return $G$

**Filtering:**
**Input:** Sets $R$ and $G$ and parameters $\tau, k$.
 1: **for** each $e \in R$ **do**
 2:   **if** $\Delta_f(e|G) < \tau$ **then**
 3:     $R = R \backslash \{e\}$.
**Output:** Return $R$.

---

**Lemma 2** *Given sets $S$ and $G$ and parameters $\tau, k$, the query complexity (i.e., the number of times that we invoke function $f$ to compute the marginal value) of Primitive* Greedy *is $O(|S|)$. Similarly, given sets $R$ and $G$ and parameters $\tau, k$, the query complexity of Primitive* Filtering *is $O(|R|)$.*

**Proof :**  For each element $e \in S$ and as long as $|G| \leq k$, we check if the marginal value of $e$ is greater than threshold $\tau$. If this is the case, we add $e$ to $G$. So, overall we invoke the function $f$ for $O(|S|)$ times. The second part is proven similarly. $\qquad \square$

**Dynamic model.**  Let $\mathcal{S}$ be a stream of insertions and deletions of elements of an underlying ground set $V$. Suppose we want to (approximately) compute a monotone submodular maximization under $k$ cardinality constraint for a monotone function $f$ which is defined for the set $V$. We define the time $t$ to be the $t^{\text{th}}$ update (i.e., insertion or deletion) of stream $\mathcal{S}$. Let $G_t$ be a solution of the underlying set $V_t$ where $V_t$ is the set of elements that are inserted up to time $t$ but not deleted. The *update time* of a dynamic algorithm $\mathcal{A}$ is the time that $\mathcal{A}$ needs to compute a solution $G_t$ of the set $V_t$ given a solution $G_{t-1}$ of the set $V_{t-1}$.

**Oblivious adversarial model.**  We work in the oblivious adversarial model as is common for analysis of randomized data structures such as universal hashing [CW77]. This model has been used

in a series of papers on dynamic maximal matching and dynamic connectivity problems: see for example [OR10, BGS15, NS13, KKM13]. The model allows the adversary to know all the elements in the set $V$ and their arrival order, as well as the algorithm to be used. However, the adversary is not aware of the random bits used by the algorithm, and so cannot choose updates adaptively in response to the randomly guided choices of the algorithm. This effectively means that we can assume that the adversary prepares the full input (inserts and deletes) before the algorithm runs.

## 1.3 Related work

Due to rapidly growing datasets, recently research has been focused on submodular optimization in the streaming and the distributed models. Very recently, Kazemi, Mitrovic, Zadimoghaddam, Lattanzi and Karbasi [KMZ$^+$19] developed $O(k)$-space (insertion-only) streaming algorithm that computes $1/2$-approximation of the optimal solution. Due to the lack of space, see references therein for more works on the streaming model.

The first distributed algorithm for the cardinality constrained submodular maximization was due to Mirrokni and Zadimoghaddam [MZ15] who gave a $0.27$-approximation in 2 rounds without duplication and a $0.545$-approximation with significant duplication of the ground set (each element being sent to $\Theta(\frac{1}{\epsilon}\log(\frac{1}{\epsilon}))$ machines). Later, Barbosa, Ene, Nguyen and Ward [dPBENW16] achieves a $(\frac{1}{2} - \epsilon)$-approximation in 2 rounds and was the first to achieve a $(1 - \frac{1}{e} - \epsilon)$ approximation in $O(\frac{1}{\epsilon})$ rounds. Both algorithms require $\Omega(\frac{1}{\epsilon})$ duplication. [dPBENW16] mentions that without duplication, the two algorithms could be implemented in $O(\frac{1}{\epsilon}\log(\frac{1}{\epsilon}))$ and $O(\frac{1}{\epsilon^2})$ rounds, respectively.

Very recently Liu and Vondrak [LV19] develop a simple thresholding algorithm that with one random partitioning of the dataset (no duplication) achieves the following: In 2 rounds of MapReduce, they obtain a $(1/2 - \epsilon)$-approximation. In 4 rounds they obtain a $5/9$ approximation. More generally, in $2t$ rounds, they achieve $(1 - (1 - \frac{1}{t+1})^t - \epsilon)$-approximation which is optimal for this type of algorithm. Their algorithm is inspired by the streaming algorithms that are presented in [KMVV15] and [MV19]. It is also similar to the algorithm of Assadi and Khanna [AK18] who study the communication complexity of the maximum coverage problem. Our dynamic algorithm is inspired by the MapReduce algorithms that Liu and Vondrak developed in [LV19].

# 2 Dynamic algorithm with expected $O(\epsilon^{-2}k \cdot \log^2 n)$ amortized update time

We first develop the offline algorithm and then in the next sections we show how to handle insertions and deletions of elements of an underlying ground set $V$.

## 2.1 Offline algorithm

Suppose we are given a ground set $V$ of size $n$ endowed with a monotone submodular function $f : 2^V \to \mathbb{R}^+$ under a cardinality constraint parameterized by $0 < k \le n$. Recall that $OPT = \max_{S \subseteq V : |S| \le k} f(S)$. Recall that we denote by $\mathcal{O}$ a subset of size at most $k$ that achieves the optimal value $OPT = f(\mathcal{O})$.

The algorithm is as follows: Suppose we are given $OPT$ and let $R_0 = V$. We recursively sample a set $S_i \subseteq R_{i-1}$ of $O(\epsilon^{-2} \log n)$ elements uniformly at random. We then set a threshold $\tau = \frac{OPT}{2k}$ and invoke the greedy algorithm (described in Section 1.2) with the input set $S_i$ to return a set $G_i$ of elements whose marginal gain is at least the threshold $\tau$. Finally, we filter those elements of $R_{i-1} \backslash S_i$ whose marginal gain with respect to the set $G_i$ is below $\tau$ and let $R_i$ be the rest of elements (i.e, those whose marginal gain with respect to the set $G_i$ is at least $\tau$). We then recurse if $R_i$ is not empty.

We first prove that the approximation ratio of Algorithm Sampling is $1/2$.

**Lemma 3** *Suppose we are given a ground set $V$ of size $n$ endowed with a monotone submodular function $f : 2^V \to \mathbb{R}^+$ under a cardinality constraint parameterized by $0 < k \le n$. Then, Algorithm Sampling returns a set $G \subseteq V$ of size at most $k$ such that $f(G) \ge \frac{1}{2} \cdot OPT$, where $OPT = \max_{S \subseteq V : |S| \le k} f(S)$.*

<div style="border:1px solid #000; padding:10px;">

**Sampling**

**Input:** A ground set $V$ of size $n = |V|$ and a parameter $0 < k \leq n$.

1: Let $s = 9\epsilon^{-2}\log n$, $\tau = \frac{OPT}{2k}$ and $i = 0$.
2: Let $R_i = V$ and $G_i = \emptyset$.
3: Invoke $\mathsf{Loop}(i, S_j, R_j, G_j)$ to return $i^*, V, S_j, R_j, G_j$ for $j \in \{0, 1, \cdots, i^*\}$.

**Output:** Return $i^*, V, S_i, R_i, G_i$ for $i \in \{0, 1, \cdots, i^*\}$.

---

**Loop:**
**Input:** $i, S_j, R_j, G_j$ for $j \leq i$.

1: **while** $R_i \neq \emptyset$ **do**
2:   Let $i \leftarrow i + 1$.
3:   Let $S_i \subseteq R_{i-1}$ be a set of elements sampled with probability $p = \min(\frac{s}{|R_{i-1}|}, 1)$.
4:   Let $G_i$ be the output of Algorithm $\mathsf{Greedy}(S_i, G_{i-1}, \tau, k)$.
5:   Let $R_i$ be the output of Algorithm $\mathsf{Filtering}(R_{i-1}\backslash S_i, G_i, \tau, k)$.
6: Let $i^*$ be last $i$ after leaving the while loop.

**Output:** Return $i^*, V, S_i, R_i, G_i$ for $i \in \{0, 1, \cdots, i^*\}$.

</div>

**Proof :**  Recall that the set $G$ contains elements whose marginal value is at least $\frac{OPT}{2k}$. We have two cases. The first case is when $|G| = k$ and the second case is when $|G| < k$. As for the first case, $f(G) \geq \frac{OPT}{2}$.

For the second case, suppose $\mathcal{O}$ is the optimal solution. Since $f$ is submodular and monotone, we then have $OPT = f(\mathcal{O}) \leq f(\mathcal{O} \cup G) \leq f(G) + \sum_{e \in \mathcal{O}\backslash G} \Delta_f(e|G) \leq f(G) + k \cdot \frac{OPT}{2k}$ .   $\square$

Next we prove that at each step $i$ of Algorithm $\mathsf{Sampling}$, we either have $|G_i| \geq |G_{i-1}| + 1$ or $|R_i| \leq \frac{|R_{i-1}|}{2}$.

**Lemma 4** *At each step $i$ of Algorithm $\mathsf{Sampling}$, with probability at least $1 - \frac{1}{n^3}$, we either have $|G_i| \geq |G_{i-1}| + 1$ or $|R_i| \leq \frac{|R_{i-1}|}{2}$.*

**Proof :**  Recall that we sample each element of the ground set $V$ with probability $p = \min(\frac{s}{|R_{i-1}|}, 1)$ where $s = 9\epsilon^{-2}\log n$. Thus, $\mathbf{E}[|S_i|] = p \cdot |R_{i-1}| = \frac{s}{|R_{i-1}|} \cdot |R_{i-1}| = s$ .

Now we use the chernoff bound to prove that the size of $S_i$ cannot be much less than its expectation with a reasonably good probability. In particular, we have

$$\mathbf{Pr}[|S_i| \leq (1-\epsilon) \cdot \mathbf{E}[|S_i|]] \leq \exp(-\epsilon^2 \cdot \mathbf{E}[|S_i|]/3) = \exp(-\epsilon^2 \cdot \frac{s}{3}) \leq 1/n^3 \ .$$

Let us condition on the event that $|S_i| \geq s/2$ for $\epsilon \leq 1/2$ that happens with probability $1 - 1/n^3$. If before step $i$, there are at least $\frac{|R_{i-1}|}{2}$ elements in $R_i$ whose marginal gain with respect to the current set $G_{i-1}$ are greater than $\frac{OPT}{2k}$, then for $\epsilon \leq 1/2$ and with probability at least $1 - (1 - \frac{1}{2})^{s/2} > 1 - 1/n^9$, we sample at least one of them and add it to $S_i$.

Thus, at Step $i$, we have two cases, either there are at least $\frac{|R_{i-1}|}{2}$ elements in $R_i$ whose marginal gain with respect to the current set $G_{i-1}$ are greater than $\frac{OPT}{2k}$ or not. If the first case happens, then with probability $1 - 1/n^9$ at least one of them is in $S_i$ which means the greedy algorithm at Step 4 of Algorithm $\mathsf{Sampling}$ will pick one of them and therefore, $|G_i| \geq |G_{i-1}| + 1$. If the second case occurs, the filtering algorithm at Step 5 of Algorithm $\mathsf{Sampling}$ shaves off those elements of $R_{i-1}$ whose marginal gain with respect to the current set $G_{i-1}$ are less than $\frac{OPT}{2k}$ and move the remaining elements to $R_i$. Thus, $|R_i| \leq |R_{i-1}|/2$.

$\square$

**Corollary 5** *With probability $1 - 1/n^3$, the number of iterations of Subroutine $\mathsf{Loop}$ in Algorithm $\mathsf{Sampling}$ is at most $i^* \leq O(k \log n)$.*

**Lemma 6** *The number of times that we query the function $f$ (i.e., query complexity) to compute the marginal value in Algorithm* Sampling *is* $O(nk \log n)$.

**Proof :** Recall that in Algorithm Sampling at each Step $i$ we invoke once Subroutine Greedy and once Subroutine Filtering. Using Lemma 2, the query complexity of either of these subroutines is linear in terms of the input size. Using Corollary 5 the loop of Algorithm Sampling iterates for $O(k \log n)$. Thus, the query complexity of Algorithm Sampling is $O(nk \log n)$. □

## 2.2 Insertion

Suppose at a time $t$ a new element $e$ is inserted. We assume that we are given the maximum iteration $i^*$ which using Corollary 5 is $i^* = O(k \log n)$ and the sets $V, S_i, R_i, G_i$ for $i \in \{0, 1, \cdots, i^*\}$ in the beginning of time $t$. We first add $e$ to the ground set $V$. Recall that at any time $t$ the ground set contains those elements that have been inserted up to the time $t$, but not deleted.

At Step $i$, with probability $p = \frac{s}{|R_{i-1}|}$ for $s = 9\epsilon^{-2} \log n$, we do a *heavy computation* and with probability $1 - p$, we do a *light computation*. During the heavy computation, we restart the sampling process (Algorithm Loop) with the input set $R_{i-1}$. As for the light computation we check if the marginal gain of adding the element $e$ to $G_i$ is above the threshold $\tau$, we then add $e$ to the set $R_i$. Otherwise, we break the for-loop and terminate Subroutine Insertion.

---

**Insertion**

**Input:** A new element $e$ and $i^*, V, S_i, R_i, G_i$ for $i \in \{0, 1, \cdots, i^*\}$.
1: Let $s = 9\epsilon^{-2} \log n$ and $\tau = \frac{OPT}{2k}$.
2: Let $V = V \cup \{e\}$ and $R_0 = V$.
3: **for** $i = 1$ to $i^*$ **do**
4:     With probability $p = \min(\frac{s}{|R_{i-1}|}, 1)$, let $o = True$.
5:     With probability $1 - p$, let $o = False$.
6:     **if** $o = True$ **then**
7:         Invoke Loop$(i, S_j, R_j, G_j)$ to return $i^*, V, S_j, R_j, G_j$ for $j \in \{0, 1, \cdots, i^*\}$.
8:         Break the FOR loop.
9:     **else**
10:        **if** $\Delta_f(e|G_i) \geq \tau$ **then**
11:           Let $R_i = R_i \cup \{e\}$.
12:        **else**
13:           Break the FOR loop.
**Output:** Return the set $G$.

---

**Lemma 7** *Suppose at a time $t$ an element $e$ is inserted. Then, the expected computation time of Subroutine* Insertion *is* $\mathbf{E}[UpdateTime(Insertion)] = c\epsilon^{-2}k^2 \log^3 n$, *where $c$ is a large enough constant and $n$ is the maximum size of the ground set $V$ at any time $t$.*

**Proof :** At Step $i$, we do a heavy computation with probability $p = \frac{s}{|R_{i-1}|}$ for $s = 9\epsilon^{-2} \log n$, and a light computation with probability $1 - p$. During the heavy computation we restart the sampling process (Algorithm Loop) with the input set $R_{i-1}$. Using Lemma Lemma 6 this takes $O(|R_{i-1}| \cdot k \log n)$ and it returns a set $G$ of size at most $k$ for which $f(G) \geq \frac{OPT}{2}$.

On the other hand, a light computation takes $O(1)$ time. Here we assume that $\Delta_f(e|G) \geq \tau$ takes constant time. Thus, $\mathbf{E}[\text{UpdateTime(Insertion)}] = \sum_{i=1}^{i^*}(\frac{s}{|R_{i-1}|} \cdot O(|R_{i-1}|k \log n) + (1 - \frac{s}{|R_{i-1}|}) \cdot O(1)) = O(\epsilon^{-2}k^2 \log^3 n)$ . □

## 2.3 Deletion

Suppose at a time $t$ a new element $e$ is deleted. We assume that we are given the maximum iteration $i^*$ which using Corollary 5 is $i^* = O(k \log n)$ and the sets $V, S_i, R_i, G_i$ for $i \in \{0, 1, \cdots, i^*\}$ in the

beginning of time $t$. We first remove $e$ from the ground set $V$. Recall that at any time $t$ the ground set contains those elements that have been inserted up to the time $t$, but not deleted.

At Step $i$ of Algorithm Deletion, if $e \in G_i$, we do a *heavy computation*. Otherwise we do a light computation. During the heavy computation, we restart the sampling process (Algorithm Loop) with the input set $R_{i-1}$. As for the light computation we check if $e \in R_i$ and if this is the case, we then remove $e$ from $R_i$. The pseudocode of the insertion subroutine is given in below.

---

**Deletion**

**Input:** An element $e$ and $i^*, V, S_i, R_i, G_i$ for $i \in \{0, 1, \cdots, i^*\}$.
1: Let $s = 9\epsilon^{-2} \log n$ and $\tau = \frac{OPT}{2k}$.
2: Let $V = V \backslash \{e\}$ and $R_0 = V$.
3: **for** $i = 1$ to $i^*$ **do**
4:    **if** $e \in G_i$ **then**
5:       Invoke Loop$(i, S_j, R_j, G_j)$ to return $i^*, V, S_j, R_j, G_j$ for $j \in \{0, 1, \cdots, i^*\}$.
6:       Break the FOR loop.
7:    **else if** $e \in R_i$ **then**
8:       $R_i = R_i \backslash \{e\}$.
**Output:** Return the set $G$.

---

**Lemma 8** *Suppose at a time $t$ an element $e$ is deleted. Then, the expected computation time of Subroutine* Insertion *is* $\mathbf{E}[\text{UpdateTime(Deletion)}] = c\epsilon^{-2}k^2 \log^3 n$, *where $c$ is a large enough constant and $n$ is the maximum size of the ground set $V$ at any time $t$.*

**Proof :** At Step $i$, we do a heavy computation if $e \in G_i$ which happens with probability $p = \frac{s}{|R_{i-1}|}$ for $s = 9\epsilon^{-2} \log n$. Otherwise we do a light computation in which we check if $e \in R_i$ and if this is the case, we then remove $e$ from $R_i$. Observe that this happens with probability less than $1 - p$.

During the heavy computation we restart the sampling process (Algorithm Loop) with the input set $R_{i-1}$. Using Lemma Lemma 6 this takes $O(|R_{i-1}| \cdot k \log n)$ and it returns a set $G$ of size at most $k$ for which $f(G) \geq \frac{OPT}{2}$.

On the other hand, a light computation takes $O(1)$ computation time. Here we assume that $\Delta_f(e|G) \geq \tau$ takes constant time. Thus, $\mathbf{E}[\text{UpdateTime(Deletion)}] = \sum_{i=1}^{i^*} \left( \frac{s}{|R_{i-1}|} \cdot O(|R_{i-1}|k \log n) + (1 - \frac{s}{|R_{i-1}|}) \cdot O(1) \right) = O(\epsilon^{-2}k^2 \log^3 n)$ . $\qquad \square$

## 3   Dynamic algorithm with high probability guarantee

Here we show how to convert a dynamic algorithm with expected amortized update time into a dynamic algorithm that with high probability has amortized update time. Suppose we have a ground set $V$ that is initialized as an empty set. Let $\mathcal{S} = \{Update(e_1), \cdots, Update(e_z)\}$ be a sequence of element updates of the underlying ground set $V$ where $Update(e_\ell)$ is either $Insert(e_\ell)$ or $Delete(e_\ell)$. Suppose that there exists a randomized dynamic algorithm $\mathcal{A}$ with two subroutines Insertion$_\mathcal{A}(e_\ell)$ and Deletion$_\mathcal{A}(e_\ell)$, where the first one inserts an element $e_\ell$ into a ground set $V$ and the second one deletes an (already inserted) element from the ground set $V$. We assume that:

1. **Efficiency:** There exists asymptotic functions $g(n)$ and $h(n)$ dependent on the size of the ground set $V$, i.e., $n = |V|$ for which the expected update times are

$$\mathbf{E}[\text{UpdateTime(Insertion}_\mathcal{A}(e_\ell))] = g(n) \quad \text{and} \quad \mathbf{E}[\text{UpdateTime(Deletion}_\mathcal{A}(e_\ell))] = h(n) .$$

2. **Quality:** For any sequence of updates (inserts and deletes), Algorithm $\mathcal{A}$ maintains an $\alpha$-approximation of a cardinality-constrained monotone submodular maximization.

As for Algorithm $\mathcal{A}$ we can use the algorithm in Section 2 whose approximation factor is $\alpha = (\frac{1}{2} - \epsilon)$. The expected insertion and deletion time of this algorithm is $O(\epsilon^{-3}k^2 \log^5 n)$. Using the algorithm

$\mathcal{A}$, we present a randomized dynamic algorithm that with probability at least $1 - \frac{\delta}{n^4}$ maintains an $\alpha$-approximation of a cardinality-constrained monotone submodular maximization for any sequence of $z$ updates (inserts and deletes) in $O((g(n) + h(n)) \cdot \log(\frac{n}{\delta}))$ amortized update time.

---

**Dynamic-Submodular-Maximization**

**Input:** A Sequence $\mathcal{S} = \{Update(e_1), \cdots, Update(e_z)\}$ of element updates of an underlying ground set $V$ where $Update(e_\ell)$ is either $Insert(e_\ell)$ or $Delete(e_\ell)$.

1: Let $V$ be a ground set that is endowed with a monotone submodular function $f : 2^V \to \mathbb{R}^+$ under a cardinality constraint parameterized by $0 < k \leq n$. Suppose $V$ is initialized to an empty set.
2: Initialize $y = 8 \log(n/\delta)$ runs $R_1, \cdots, R_y$ in parallel.
3: **for** $R_r$ where $r \in [y]$ in parallel **do**
4:      Invoke Algorithm $\mathcal{A}$.
5:      **for** each $Update(e_\ell)$ **do**
6:         **if** $Update(e_\ell)$ is $Insert(e_\ell)$ **then**
7:            Invoke $\mathsf{Insertion}_{\mathcal{A}}(e_\ell)$.
8:         **else**
9:            Invoke $\mathsf{Deletion}_{\mathcal{A}}(e_\ell)$.
10:      **if** the update time of $R_r$ up to now is greater than $3cz \cdot \log^3 n \; 3z \cdot (g(n) + h(n)) \cdot \log(\frac{n}{\delta})$ **then**
11:         Stop the run $R_r$.

**Output:** At any time $t \in [z]$, report the set $S$ whose $f(S)$ is the median of the sets returned by those runs $R_r$ that are survived.

---

**Theorem 9** *Let $0 < \delta < 1$ be a parameter. Then, there is a randomized algorithm that with probability at least $1 - \delta/n^2$, maintains an $\alpha$-approximation of a cardinality-constrained monotone submodular maximization in time $O((g(n) + h(n)) \cdot z \cdot \log(\frac{n}{\delta}))$. That is, the amortized update time of this algorithm is $O((g(n) + h(n)) \cdot \log(\frac{n}{\delta}))$.*

**Proof :** We define $z$ random variables $X_1, \cdots, X_z$ corresponding to the $z$ updates where $X_\ell$ corresponds to the update time of the element $e_\ell$. We have

$$\mathbf{E}[X_\ell] \leq \mathbf{E}[UpdateTime(Insertion_{\mathcal{A}}(e_\ell))] + \mathbf{E}[UpdateTime(Deletion_{\mathcal{A}}(e_\ell))] \leq g(n) + h(n) \ .$$

Let $X = \sum_{\ell=1}^z X_\ell$. We then have $\mathbf{E}[X] \leq z \cdot (g(n) + h(n))$. Using Markov Inequality, $\mathbf{Pr}[X \geq 3z \cdot (g(n) + h(n))] \leq 1/3$ . Next we increase the probability of correctness to $1 - \delta/n^3$. For the sequence $\mathcal{S} = \{Update(e_1), \cdots, Update(e_z)\}$ element updates of the underlying ground set $V$, we run $y = 8 \log(n/\delta)$ instances of Algorithm $\mathcal{A}$ in parallel. Let $R_1, \cdots, R_y$ be the set of these $y$ runs. At any time $1 \leq t \leq z$, if we observe that for a run $R_r$ the sum of the update times from the beginning of the sequence $\mathcal{S}$ up to the time $t$ is greater than $3z \cdot (g(n) + h(n))$, we stop the run $R_r$.

Let $Y_r$ corresponds to the run $R_r$ such that $Y_r = 1$ if for the $r^{\text{th}}$ run the total update time of $R_r$ from the time $1$ to $t$ is greater than $3z \cdot (g(n) + h(n))$, and $Y_r = 0$ otherwise. Therefore, $\mathbf{E}[Y_r] = p \leq 1/3$. Let $a = 1/2$ and $Y = \sum_{r=1}^y Y_r$. Now we can use additive Chernoff Bound 10.

**Lemma 10 (Additive Chernoff Bound)** *[Che52] Let $Y_1, \cdots, Y_m$ denote $m$ identically distributed and independent random variables such that $\mathbf{E}[Y_i] = p$ for $1 \leq i \leq n$ for a fixed $0 \leq p \leq 1$. Let $0 < t < 1, t \geq p$. For $Y = \sum_{i=1}^m Y_i$ it holds that $\mathbf{Pr}[Y \geq t \cdot m] \leq ((\frac{p}{t})^t \cdot (\frac{1-p}{1-t})^{(1-t)})^m$.*

Thus, we have $\mathbf{Pr}[Y \geq y/2] \leq ((\frac{p}{a})^a \cdot (\frac{1-p}{1-a})^{1-a})^y \leq (\sqrt{2/3} \cdot \sqrt{\frac{2/3}{1/2}})^y \leq (\sqrt{8/9})^y \leq \delta/n^8$ , for $y \geq 8 \log_{\sqrt{9/8}}(n/\delta)$. By the relation between logarithms we then have $y \geq 8 \log(n/\delta)$. We assume that $z \leq n(n+1)/2 = n^2/2 + n/2 \leq n^2$. After every $n^2$ updates we can re-run Algorithm Dynamic-Submodular-Maximization from scratch. Therefore, using a union bound, with probability at least $1 - \delta/n^2$ after every update there exist at least $y/2$ runs that survive. At any time $t \in [z]$, we report the set $S$ whose $f(S)$ is the median of the sets returned by those runs $R_r$ that are survived. $\qquad \square$

## Acknowledgements

The work of Morteza Monemizadeh was partially supported by Department of Mathematics and Computer Science, TU Eindhoven, the Netherlands.

## Broader Impact

In this paper we introduced a simple but elegant dynamic algorithm for monotone submodular functions under a cardinality constraint. Submodular functions have plenty of applications in machine learning and combinatorial optimization and we believe researchers in both these areas would benefit from this simple algorithm. Moreover we believe that the simplicity of this algorithm has an educational impact. In particular, we think it can be used as a textbook example to explain the growing area of dynamic algorithms for undergraduate and graduate students.

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
