[Supplementary Material]

# Dynamic Submodular Maximization
# (Supplementary Material)

**Morteza Monemizadeh**
Department of Mathematics and Computer Science
TU Eindhoven, the Netherlands
m.monemizadeh@tue.nl

## 1   Dynamic Algorithm with $\tilde{O}(\sqrt{kn})$ Amortized Update Time

In this section we prove the following theorem.

**Theorem 1** *Suppose we start with an empty set $V$. Then, there exists a randomized dynamic algorithm that with probability at least $1 - \frac{1}{n^2}$ maintains a $\frac{1}{2}$-approximation of a cardinality-constrained monotone submodular maximization for any sequence of $z$ updates (inserts and deletes) in $\tilde{O}(z \cdot \sqrt{kn})$[1] time, i.e., $\tilde{O}(\sqrt{kn})$ amortized update time.*

**Overview of Proof of Theorem 1.**    An interesting property of a submodular function $f : 2^V \to \mathbb{R}^+$ is that it satisfies $f(A \cup \{e\}) - f(A) \geq f(A \cup \{e\}) - f(A)$ for all $A \subseteq B \subseteq V$ and $e \notin B$. Our main idea is to combine this property with a logarithmic rate of sampling and then greedily collect the heavy items (whose marginal gain are above a threshold) and remove light items (whose marginal gain are below a threshold) at each rate.

Let us first consider the offline scenario. We later show how to handle insertion and deletion of elements. Suppose we are given a ground set $V$ of size $n$ endowed with a monotone submodular function $f : 2^V \to \mathbb{R}^+$ under a cardinality constraint parameterized by $0 < k \leq n$. Recall that $OPT = \max_{S \subseteq V : |S| \leq k} f(S)$. Recall that we denote by $S^*$ the subset of size at most $k$ that achieves the optimal value $OPT = f(S^*)$. Our algorithm (that we call it Algorithm Sampling) is a refinement of Algorithm [LV19]. It returns a set $G \subseteq V$ of size at most $k$ with approximation guarantee $f(G) \geq \frac{1}{2} \cdot OPT$. We prove it in this section for the sake of completeness.

The algorithm is as follows: Suppose we are given the value of $OPT$. We sample a set $S \subseteq V$ of $O(\sqrt{kn})$ elements uniformly at random. We set a threshold $\tau = \frac{OPT}{2k}$ and invoke the greedy algorithm (described in main part of the paper) with the input set $S$ to return a set $G_S$ of elements whose marginal gain is at least the threshold $\tau$. We then filter (described in main part of the paper) those elements of $V \backslash S$ whose marginal gain with respect to the set $G_S$ is below $\tau$. Observe that the set $G_S$ might be empty. Let $R$ be the set of points that survive after the filtering step. We prove that the size of $R$ drops significantly, that is, $|R| \leq O(k \cdot \sqrt{n})$. We later invoke the greedy algorithm on the input set $R$ to complete the set $G$ of elements whose marginal gain is at least the threshold $\tau$.

Next consider a dynamic scenario where elements are inserted to $V$ or deleted from $V$. Suppose at a time $t$ a new element $e$ is inserted. We assume that we are given the sets $V, S, G_S, R, G$ in the beginning of time $t$. We first add $e$ to the ground set $V$. Recall that at any time $t$ the ground set contains those elements that have been inserted up to the time $t$, but not deleted.

With probability $p = 4\sqrt{\frac{k}{n}}$, we do a *heavy computation*. During the heavy computation, we restart the sampling process (Algorithm Sampling) from scratch. With probability $1 - p$, we do a *light computation*. A light computation consists of checking two conditions.

- If the marginal gain of adding the element $e$ to $G_S$ is above the threshold $\tau$ and $G_S$ has less than $k$ elements, we add $e$ to the set $R$.

- If the first condition is fulfilled and the marginal gain of adding the element $e$ to $G$ is above the threshold $\tau$ and $G$ has less than $k$ elements, we add $e$ to the set $G$.

Next suppose at a time $t$ a new element $e$ is deleted. We assume that we are given the sets $V, S, G_S, R, G$ in the beginning of time $t$. We first remove $e$ from the ground set $V$. Recall that at any time $t$ the ground set contains those elements that have been inserted up to the time $t$, but not deleted.

If $e \in G_S$, we do a *heavy computation*. During the heavy computation, we restart the sampling process (Algorithm Sampling) from scratch. If $e \notin G_S$, we do a *semi-heavy computation*. As for the semi-heavy computation, if $e \notin R$, we do nothing. But if $e \in R$, we eliminate $e$ from $R$ and then run the greedy algorithm for the set $R$ to find those elements of $R$ whose marginal gain with respect to the set $G_S$ is at least the threshold $\tau$.

To have the dynamic algorithm that works with high probability we create $O(\log n)$ instances of this recursive sampling and run all of them in parallel. After any sequence of $z$ insertions and deletions, we drop those instances whose computations are more than $cz\sqrt{kn}$ for some constant $c$. We show that with high probability it remains at least one instance whose total computation is at most $cz$ $sqrtkn$. That is, the amortized update time of that instance is $c\sqrt{kn}$.

Finally we should mention that for the threshold $\tau$ we choose $\frac{OPT}{2k}$ assuming we know $OPT$. In reality we do not know $OPT$. We can consider two scenarios. The first scenario is when we are given a bound on the maximum element of $V$, that is, say $\max_{e \in V} f(e) = \Theta(\Gamma)$. This is actually a fair assumption that we often make when we generalize the insertion-only streaming model to dynamic streaming models. For example, Frahling and Sohler in [FS05] show that we can find coresets of small size for many clustering problems (a subset of submodular optimization problems) in dynamic geometric streams if we have an upper-bound on the maximum cost of the optimal clustering, something which is not possible if we do not have such an upper-bound. Since $OPT \leq ck \cdot \Gamma$ for a reasonably large constant $c$, we run our recursive sampling algorithm for $\ell \in [0..\epsilon^{-1} \cdot \log(ck\Gamma)]$ guesses $(1 + \epsilon)^\ell$ of $OPT$ and report the best solution of all guesses. This blows up the update time by a factor $\epsilon^{-1} \log(k\Gamma)$ and the approximation factor would be $(1/2 - \epsilon)$.

If we are not given such a bound, we can keep a max heap of the elements that are inserted but not deleted at any time $t$. The insertion and deletion times of the max heap are logarithmic in terms of the number $n$ of items that are stored in the max heap. Finding the maximum $r$ elements stored in the heap can be done in $O(r \log n)$ time. We then do as follows. We sample a set $S$ of $O(\sqrt{nk})$ elements and let $\Gamma = \max_{e \in V} f(e)$ and run the algorithm as for the first scenario. In parallel, at any time $t$, we extract the set $T$ of maximum $O(\sqrt{nk})$ elements from the max heap and run the greedy algorithm due to Nemhauser, Wolsey and Fisher [NWF78] for $T$. At the end, we report the best solution of these two parallel runs. In this way, the update time remains as $\tilde{O}(\sqrt{nk})$ and the approximation factor would be $(1/2 \pm \epsilon)$.

We first give the offline algorithm and prove that the approximation factor of this algorithm is $(1/2 \pm \epsilon)$. Later we show how we can implement the insertion and deletion subroutines in $\tilde{O}(\sqrt{kn})$ amortized update time.

## 1.1 Offline Algorithm

We first prove the size of the set $R$ is at most $\sqrt{nk}$ with high probability and then we prove that the approximation ratio of Algorithm Sampling is $1/2$.

**Lemma 2** *The size of the set $R$ returned by Subroutine Filtering in Step $4$ of Algorithm Sampling is at most $\sqrt{nk}$ with probability at least $1 - e^{-\frac{k}{12}}$.*

**Proof :** Recall that we sample each element of the ground set $V$ with probability $p = 4\sqrt{\frac{k}{n}}$. Thus,

$$\mathbf{E}[|S|] = p \cdot |V| = p \cdot n = 4\sqrt{\frac{k}{n}} \cdot n = 4\sqrt{kn} \ .$$

> **Sampling**
>
> **Input:** A ground set $V$ of size $n = |V|$ and a parameter $0 < k \le n$.
> 1: Let $\tau = \frac{OPT}{2k}$.
>
> 2: Let $S \subseteq V$ be a sample set where each $e \in V$ is sampled with $p = 4\sqrt{\frac{k}{n}}$.
>
> 3: Let $G_S$ be the output of Algorithm Greedy$(S, \emptyset, \tau, k)$.
> 4: Let $R$ be the output of Algorithm Filtering$(V \backslash S_i, G_S, \tau, k)$.
> 5: Let $G$ be the output of Algorithm Greedy$(R, G_S, \tau, k)$.
>
> **Output:** Return the sets $V, S, G_S, R, G$.

Now we use the multiplicative Chernoff bound to prove that the size of $S$ cannot be less than its expectation with a reasonably good probability. In particular, for $\epsilon = 1/4$ and since $n \ge k$ we have

$$\mathbf{Pr}[|S| \le (1 - \epsilon) \cdot \mathbf{E}[|S|]] \le \exp(-\epsilon^2 \cdot \mathbf{E}[|S|]/3) = \exp(-\frac{1}{16} \cdot \frac{4\sqrt{kn}}{3}) \le e^{-\frac{k}{12}} \ .$$

Let us condition on the event that $|S| \ge 3\sqrt{kn}$ that happens with probability $1 - e^{-\frac{k}{12}}$. We split the sampled set $S$ into $3k$ chunks of equal size $\sqrt{\frac{n}{k}}$. Let us call them $S_1, \cdots, S_{3k}$. We process each chunk sequentially. Suppose we want to sample the chunk $S_i$ independently one at a time. Suppose for the sets $S_1, \cdots, S_{i-1}$ we have picked a set $G$ of elements whose marginal gain are greater than $\frac{OPT}{2k}$. Now if before chunk $S_i$ there are at least $\sqrt{nk}$ elements whose marginal gain with respect to the current set $G$ are greater than $\frac{OPT}{2k}$, then with probability at least $1 - (1 - \sqrt{\frac{n}{k}})^{\sqrt{\frac{n}{k}}}$, we sample one of them and will add to $G$. This happens conditioned on any prior history of the algorithm. Thus, we can use martingale argument to bound the number of elements whose marginal gain are greater than $\frac{OPT}{2k}$ and are added to $G$. Let us define an indicator random variable for the event that at least one such an element with big marginal gain is sampled in $S_i$. So, we have $\mathbf{E}[X_i | X_1, \cdots, X_{i-1}] \ge 1/2$. Now corresponding to the random variable $X_i$, we define the random variable $Y_i = \sum_{i=1}^{i}(X_i - \frac{1}{2})$. Observe that the sequence $Y_1, Y_2, \cdots$ is a submartingale since $\mathbf{E}[Y_i | Y_1, \cdots, Y_{i-1}] \ge Y_{i-1}$. Moreover, $|Y_i - Y_{i-1}| \le 1$.

Thus, we can apply Azuma's inequality to obtain:

$$\mathbf{Pr}[Y_{3k} < -\frac{1}{2} \cdot k] \le \exp(-(\frac{1}{2} \cdot k)^2/2) = e^{\frac{k^2}{8}} \ .$$

This essentially means that with probability at least $1 - e^{\frac{k^2}{8}}$ we have $\sum_{j=1}^{3k} X_j = Y_{3k} + \frac{3}{2} \cdot k \ge k$. Thus, $|G| \ge k$. Otherwise, the number of remaining elements of marginal value at least $\frac{OPT}{2k}$ drops below $\sqrt{nk}$. $\qquad\square$

**Lemma 3** *Suppose we are given a ground set $V$ of size $n$ endowed with a monotone submodular function $f : 2^V \to \mathbb{R}^+$ under a cardinality constraint parameterized by $0 < k \le n$. Then, Algorithm* Sampling-Max-k-Submodular *returns a set $G \subseteq V$ of size at most $k$ such that $f(G) \ge \frac{1}{2} \cdot OPT$, where $OPT = \max_{S \subseteq V : |S| \le k} f(S)$.*

**Proof :** Recall that the set $G$ contains elements whose marginal value is at least $\frac{OPT}{2k}$. We have two cases. The first case is when $|G| = k$ and the second case is when $|G| < k$. In the former case, $f(G) \ge \frac{OPT}{2}$.

In the latter case, suppose $\mathcal{O}$ is the optimal solution. Since $f$ is submodular and monotone, we then have:

$$OPT = f(\mathcal{O}) \le f(\mathcal{O} \cup G) \le f(G) + \sum_{e \in \mathcal{O} \backslash G} \Delta_f(e|G) \le f(G) + k \cdot \frac{OPT}{2k} \ .$$

Therefore, $f(G) \ge \frac{OPT}{2}$. $\qquad\square$

**Lemma 4** *The number of times that we query the function $f$ (i.e., query complexity) to compute the marginal value in Algorithm* Sampling *is $O(n)$.*

**Proof :**    Recall that Algorithm Sampling invokes two times Greedy and once Filtering. The query complexity of both subroutines is linear in terms of input size. Thus, the query complexity of Algorithm Sampling is $O(n)$.

<div style="text-align: right">□</div>

### 1.2   Insertion

Suppose at a time $t$ a new element $e$ is inserted. We assume that we are given the sets $V, S, G_S, R, G$ in the beginning of time $t$. We first add $e$ to the ground set $V$. Recall that at any time $t$ the ground set contains those elements that have been inserted up to the time $t$, but not deleted.

With probability $p = 4\sqrt{\frac{k}{n}}$, we do a *heavy computation*. During the heavy computation, we restart the sampling process (Algorithm Sampling) from scratch. With probability $1 - p$, we do a *light computation*. A light computation consists of checking two conditions.

- If the marginal gain of adding the element $e$ to $G_S$ is above the threshold $\tau$ and $G_S$ has less than $k$ elements, we add $e$ to the set $R$.

- If the first condition is fulfilled and the marginal gain of adding the element $e$ to $G$ is above the threshold $\tau$ and $G$ has less than $k$ elements, we add $e$ to the set $G$.

The pseudocode of the insertion subroutine is given in below.

---
**Insertion**

**Input:** A new element $e$, the sets $V, S, G_S, R, G$ and the threshold $\tau$.
1:  Let $V = V \cup \{e\}$.
2:  With probability $p = 4\sqrt{\frac{k}{n}}$, let $o = True$, and with probability $1 - p$, let $o = False$.
3:  **if** $o = True$ **then**
4:      Invoke Sampling$(V, k)$ that returns new sets $V, S, G_S, R, G$.
5:  **else**
6:      **if** $\Delta_f(e|G_S) \geq \tau$ and $|G_S| < k$  **then**
7:          $R = R \cup \{e\}$.
8:      **if** $\Delta_f(e|G) \geq \tau$ and $|G| < k$  **then**
9:          $G = G \cup \{e\}$.
**Output:** Return the updated sets $V, S, G_S, R, G$.

---

**Lemma 5** *Suppose at a time $t$ an element $e$ is inserted. Then, the expected computation time of Subroutine* Insertion *is* $\mathbf{E}[UpdateTime(Insertion)] = c \cdot \sqrt{nk}$, *where $c$ is a large enough constant and $n$ is the maximum size of the ground set $V$ at any time $t$.*

**Proof :**    We do a heavy computation with probability $p = 4\sqrt{\frac{k}{n}}$ and a light computation with probability $1 - p$. During the heavy computation we invoke Algorithm Sampling that in time $O(n)$ (see Lemma 4) returns a set $G$ of size at most $k$ for which $f(G) \geq \frac{OPT}{2}$. On the other hand, a light computation (i.e., Steps $6 - 9$ of Algorithm Sampling) needs $O(1)$ computation time. Here we assume that $\Delta_f(e|G) \geq \tau$ takes constant time. Thus,

$$\mathbf{E}[UpdateTime(Insertion)] = p \cdot O(n) + (1-p) \cdot O(1) = 4\sqrt{\frac{k}{n}} \cdot O(n) + (1 - 4\sqrt{\frac{k}{n}}) \cdot O(1) \leq 5\sqrt{nk} \ .$$

<div style="text-align: right">□</div>

## 1.3 Deletion

Suppose at a time $t$ a new element $e$ is deleted. We assume that we are given the sets $V, S, G_S, R, G$ in the beginning of time $t$. We first remove $e$ from the ground set $V$. Recall that at any time $t$ the ground set contains those elements that have been inserted up to the time $t$, but not deleted.

If $e \in G_S$, we do a *heavy computation*. During the heavy computation, we restart the sampling process (Algorithm Sampling) from scratch. If $e \notin G_S$, we do a *semi-heavy computation*. As for the semi-heavy computation, if $e \notin R$, we do nothing. But if $e \in R$, we eliminate $e$ from $R$ and then run the greedy algorithm for the set $R$ to find those elements of $R$ whose marginal gain with respect to the set $G_S$ is at least the threshold $\tau$.

The pseudocode of the insertion subroutine is given in below.

---

**Deletion**

**Input:** An element $e$, the sets $V, S, G_S, R, G$ and the threshold $\tau$.
1: Let $V = V \backslash \{e\}$.
2: **if** $e \in G_S$ **then**
3:     Invoke Sampling$(V, k)$ that returns new sets $V, S, G_S, R, G$.
4: **else**
5:     **if** $e \in R$ **then**
6:         $R = R \backslash \{e\}$.
7:         Let $G$ be the output of Algorithm Greedy$(R, G_S, \tau, k)$.
**Output:** Return the updated sets $V, S, G_S, R, G$.

---

**Lemma 6** *Suppose at a time $t$ an element $e$ is deleted. Then, the expected computation time of Subroutine* Deletion *is* $\mathbf{E}[UpdateTime(Deletion)] = c \cdot \sqrt{nk}$, *where $c$ is a large enough constant and $n$ is the maximum size of the ground set $V$ at any time $t$.*

**Proof :** We do a heavy computation if $e \in S$ which happens with probability $p = 4\sqrt{\frac{k}{n}}$. We do a semi-heavy computation if $e \notin S$ which happens with probability $1 - p$. During the heavy computation we invoke Algorithm Sampling that in time $O(n)$ (see Lemma 4) returns a set $G$ of size at most $k$ for which $f(G) \geq \frac{OPT}{2}$. On the other hand, as for a semi-heavy computation (i.e., Steps $5 - 7$ of Algorithm Sampling) if $e \in R$, we run the greedy algorithm for the set $R$ to find those elements of $R$ whose marginal gain with respect to the set $G_S$ is at least the threshold $\tau$.

From Lemma 2, with probability at least $1 - e^{-\frac{k}{12}}$, the size of the set $R$ is at most $\sqrt{nk}$ for which we need $O(\sqrt{nk})$ calls to the function $f$. Thus,

$$\mathbf{E}[UpdateTime(Deletion)] = p \cdot O(n) + (1-p) \cdot O(\sqrt{nk}) = 4\sqrt{\frac{k}{n}} \cdot O(n) + (1 - 4\sqrt{\frac{k}{n}}) \cdot O(\sqrt{nk}) \leq 5\sqrt{nk} \ .$$

$\square$

## Footnotes

[1] $\tilde{O}(g(n)) = O(\epsilon^{-1} \log n \cdot g(n))$ for an asymptotic function $g(n)$.