[Reviews · NeurIPS 2020]

Review 1

Summary and Contributions: This paper studies the problem of monotone submodular maximization under a cardinality constraint in a dynamic setting with inserts and deletes of elements. The authors give an algorithm that maintains a 1/2 - eps approximation in O(k eps^{-3} log^5 n) amortized time. The main contribution is that this amortized update time is independent of the number of deletions d, whereas prior work for this problem has update time that is linear in d. Post rebuttal: The discussion for the update time and the memory constraint addressed some of the weakness I raised about the comparison with prior work in the streaming setting. Accordingly, I revised my score from 5 to 6 and I encourage the authors to include some of this discussion in the next version of the paper.

Strengths: - First algorithm for dynamic submodular maximization with amortized update time that is independent of the number of deletions. - Interesting and non-trivial techniques

Weaknesses: - There isn't a clear description of the model and problem studied. - A main difference with previous dynamic streaming results is that previous algorithms are in the streaming setting and use a small amount of memory that is sublinear in the number of elements. This algorithm keeps all elements in memory. This difference with the previous results mentioned in this paper should be discussed. - The presentation and the writing could be improved. Especially the introduction, which could use more motivation and background.

Correctness: I have not found any errors

Clarity: Several aspects could be improved, especially: - more motivation and background before stating the main result - a clear presentation of the model before overview of proof, preliminaries, and basic primitives

Relation to Prior Work: Not sufficiently, the authors mention the results from KZK18 and MKK17, but the differences between the setting in this paper and in KZK18, MKK17, and prior work in general is not discussed. The claim that this paper proposes “the first dynamic algorithm for this problem” is an overstatement, KZK18 and MKK17 also propose dynamic algorithms.

Reproducibility: Yes

Additional Feedback: In lines 63 and 116, there is a typo in the definition of sub modularity, it should be f(A \cup e) - f(A) >= f(B \cup e) - f(B)


Review 2

Summary and Contributions: In the motonote submodular maximization problem with a cardinality constraint, given a set V, constraint k, and a submodular and monotone function f defined on subsets of V, the goal is to find a subset S of V of size at most k that (approximately) maximizes f(S). Contribution 1: algorithm The paper presents a dynamic algorithm that maintains a (1/2-eps)-approximate solution to the problem under insertions and deletions of elements of V. The amortized time of processing each insertion/deletion is O(k^2 / eps^3 log^5 n) (with high probability). The algorithm is based on a simple filtering and subsampling idea. At a high level it repeats the following loop: * sample a small number (i.e. polylog) of not-yet-chosen elements and greedily try adding them to the solution, if the marginal gain is at least OPT / (2k) * remove all not-yet-chosen elements with marginal gain OPT / (2k) The main insight in turning this to an incremental algorithm is the fact that an inserted element does not affect the result, until it's sampled, which likely does not happen until the number of not-yet-chosen elements is very small. A similar argument holds for deletions.

Strengths: * Simple and elegant algorithm * Update time independent of the size of |V|

Weaknesses: * the dependence on k in the running time is super-linear, which may prevent the algorithm from being very efficient in practice * How limiting is the assumption that f is monotone? In particular, was it used in the related results discussed in the introduction?

Correctness: Having understood the main idea, the correctness is quite clear to me. I haven't verified the running time bounds, but they seem roughly correct.

Clarity: The writing style is decent, though there is a fair amount of small errors, including incorrectly stated bound in the abstract.

Relation to Prior Work: Yes

Reproducibility: Yes

Additional Feedback: line 14: should 1/2+-eps be just 1/2-eps? line 18: k -> k^2 lines 20-27: add spaces before citations (e.g. on line 23, space between clustering and [DF07]) line 121: "e \not \in V\setminus B" seems unnecessary line 148: please define V_t before it's used Filtering algorithm, line 2: I don't think "|G| < k" is needed line 195: do you mean R_i instead of R? line 250: remove 'computation' (unnecessary) section 3: at a conceptual level, the method of getting a worst case bound is similar to what is done in https://arxiv.org/pdf/1810.10932.pdf. I've also seen this idea in other paper, so it deserves at least one citation. UPDATE: following the reviewers discussion it was decided that the faster algorithm described in the authors' response should not be taken into account in judging the paper, since a similar result has been published on arXiv recently. Hence, I keep my score unchanged.


Review 3

Summary and Contributions: This paper considers the submodular maximization under a cardinality constraint. The problem setting is dynamic with a stream of inserts and deletes of elements. The proposed algorithm randomly samples the inserted item or deleted item and decide to run Loop. Loop algorithm checks the marginal gain of each item, whether the margin is higher than a threshold or not. The proposed algorithm guarantees 1/2 optimality.

Strengths: The proposed algorithm can guarantee 1/2 optimality. This is a good result for the dynamic setting of the submodular optimization. The authors show that the proposed algorithm is computationally efficient.

Weaknesses: In general, streaming algorithms mainly consider memory constraints, while this paper does not discuss the memory efficiency of the proposed algorithm. It thus is hard to compare the proposed algorithm with other existing streaming submodular optimization algorithms. The algorithm has to know the OPT value in order to guarantee 1/2 approximation. In general, the OPT value is not given. The authors have to discuss when the algorithm does not know the OPT value and provide an alternative algorithm without the OPT information. ====================== I really appreciate the authors' responses. I still believe the assumption knowing OPT in the main algorithm is strong.

Correctness: The results make sense. I didn't check the proof in the appendix.

Clarity: Overall, this paper is easy to follow, although the motivation of the problem is not enough.

Relation to Prior Work: They have to discuss why the amortized update time is more important than the memory constraint that was the main concern of existing works.

Reproducibility: Yes

Additional Feedback:


Review 4

Summary and Contributions: In this paper, the authors study the following constrained submodular optimization problem. Given a submodular function f with ground set V and a parameter k, the goal is to compute a subset S of V with at most k elements that that maximizes the value of f(S) among all subsets of V of size at most k. This is a classical problem that has been studied in many settings and has several applications. In this paper, the authors propose a dynamic algorithm for this problem. That is, they start with an empty set V, and allow insertions and deletions into this set, while maintaining a (1/2 - epsilon)-approximation of the solution of the problem. This bound is tight in the streaming model. Their proposed algorithm has an amortized update time of O(k^2 epsilon^-3 log^5 n), where n is the max size of V throughout, and maintains the desired approximation with high probability (at least 1-1/n^2). Their algorithm combines a clever greedy strategy combined with a random sampling. Intuitively, they first propose an algorithm for the static case that samples elements at random and keeps those producing a good marginal gain, moreover, among the remaining elements, they prune away those with low marginal gain. They repeat this process sampling only from the remaining elements, which yields an algorithm that after O(k log n) rounds computes a set with high marginal gain. The definition of high and low marginal gain has to be chosen carefully to yield the desired solution. Later they show how to deal with updates. For insertions, they go through the execution of the static procedure, and they toss a biased coin on each of its rounds to see what would have happened if the new element would have been sampled, if it would have, then they re-run the algorithm and recreate this scenario. While this is expensive, it happens with relatively small probability. After computing the expected running time, they arrived to the proposed update time. Additional details are needed to obtain the result with high probability. The deletions are handled in a similar way, although extra details are needed, and it becomes the hardest case.

Strengths: The authors offer what they claim is the first dynamic algorithm for the studied problem. This is almost true. as it certainly improves upon previous algorithms published before the submission deadline of NeurIPS. Namely, it removes the dependency that previous algorithms had on the number of deletions, which turns to be the main contribution of this paper. Since submodular optimization is an important topic in NeurIPS, I believe this is a relevant result for the NeurIPS community.

Weaknesses: -My only concern with this paper is that in recent weeks (couple of days after the submission of NeurIPS), a paper appeared in arXiv solving the same problem and obtaining what they claim is a better running time: "Fully Dynamic Algorithm for Constrained Submodular Optimization". Their claimed update time is O(epsilon^-3 log(k)^5 log^2 n), which improves the dependency on both n and k. I cannot claim to have read this other paper in detail, but the authors might need to compare with it and point out their differences and similarities. -Usually the running time of these algorithms is measured on the number of oracle calls to the function f. My understanding is that this is also what they use in this paper, but they never explain it explicitly. Since evaluating f could be expensive, this is relevant for many applications.

Correctness: I reviewed most of the proofs of the paper and I believe the results are correct. The running time dependencies and the computation of the expected running time are correct in my opinion.

Clarity: The paper is nicely written. They chose to first present an overview of the results and the techniques used, and later get into the details by going again over all the previously discussed steps. This allows the reader to get familiar with the whole strategy before going into details.

Relation to Prior Work: The related work section is kept a bit short, but the main discussion of related algorithms is moved to other parts of the paper. I think it is complete, but I would like to see more mentions of lower bounds in the approximation factor, why is it that 1/2 - epsilon is the best that you can achieve with your algorithm? Also, as mentioned above, now there is the need to include the results of the other paper "Fully Dynamic Algorithm for Constrained Submodular Optimization".

Reproducibility: Yes

Additional Feedback: Small comments: Line 63: Typo in definition of sub-modularity (missing B). Line 73: You could mention something about tau before using it out of the blue. -you could define more carefully what is a marginal gain before using it. ----------------EDIT-------------------- After reading the rebuttal and having a discussion with the other reviewers, we agree that the paper "Fully Dynamic Algorithm for Constrained Submodular Optimization" should not be used to assess the strength of this submission as it was published after the Neurips submission according to the guidelines. Other than that, I would have liked to see a clearer explanation of the 1/2 lower bound does not apply here, and why they believe that they can do better than 1/2 for this specific problem. However, this does not affect my opinion and I keep my score of acceptance.


Review 5

Summary and Contributions: This paper studies online submodular optimization with deletions.

Strengths: /

Weaknesses: /

Correctness: /

Clarity: /

Relation to Prior Work: /

Reproducibility: Yes

Additional Feedback: The dependence on the value OPT is a weakness of the algorithm in this paper. The authors discuss the worry about needing OPT it in the paper (lines 95-113) and attempt to fix it. On the other hand, the paper seems to have a bug in this procedure for dealing with getting an estimate of OPT. The basic idea of BV14 and KDD-14 is that when you don't know OPT, but you have a lower and upper bound on it, you can run the algorithm multiple times using estimates of OPT, and then take the best solution you find, and when done this leads to a guarantee. For cardinality constrained monotone submodular max, a simple lower bound on OPT is \max_v f(v) and a simple upper bound is k*\max_v f(v), i.e., \max_v f(v) <= OPT <= k*\max_v f(v) Given this range, we can try running the algorithm with estimates of OPT taking the form (1+\epsilon)^\ell for l \in Z and l can be positive or negative. I.e., we start with \ell such that (1+\epsilon)^\ell = max_v f(v) and end with \ell such that (1+\epsilon)^\ell = k*max_v f(v) and there are O(\log(k)/\epsilon) such estimates that must be tested, so this increases the time by a factor of O(\log(k)/\epsilon) (or you can run them in parallel; either way some resource, either time, or compute&memory, increases by a factor of O(\log(k)/\epsilon)). In the present paper, they say they can do this only with an upper bound on \Gamma >= max_v f(v) (line 97 of the paper). On line 103 of the they give a range of values of \ell to try, of the form $[0, \dots, \epsilon^{-1} \log(k \Gamma)]$ which means that their complexity increases by a factor $\epsilon^{-1} \log(k \Gamma)$ as they say on line 104. However, I believe this is wrong, since if \Gamma > \max_v f(v) then where do you start? Since $(1+\epsilon)^\ell > 1$ for $\epsilon > 0$ and $\ell > 0$, so you will eventually hit any upper bound $\Gamma$ given so this is ok. Without a lower bound, however, you don¹t know how small the submodular function valuations are, so if you start from 0 and use the range $[0, \dots, \epsilon^{-1} \log(k \Gamma)]$, you can indeed do that with $(1+\epsilon)^\ell$ with negative $\ell$ (which is what the BV14 and KDD-14 papers allow) and you can get as small as you want since as $(1+\epsilon)^\ell$ -> 0 as $\ell -> -\infty$ but then how negative should $\ell$ get? It would be unbounded from below, and without having a lower bound on OPT, therefore, would imply a trivial infinite increase in computational cost. It seems they get the $\log(k\Gamma)/\epsilon$ factor simply just by equating $(1+\epsilon)^\ell = k*\Gamma$ and then using only the upper bound on OPT, but they ignore the lower bound (like BV14 and KDD-14). If we had a lower bound, we could get an increase in complexity of $\log(k)/\epsilon$ which could be better or worse than what they give depending on if $\Gamma > 1$ or $\Gamma < 1$. They also do talk (on lines 105 - 113) about a heap approach as well that they say (lines 112-113) it is described the supplementary but the description they give is very vague. They do state that they can set $\Gamma = \max_v f(v)$ in the paragraph on the heap approach, and if we could get the value of $\max_v f(v)$, we could do the lower bound and fix the above problem, but given only oracle access to $f$, this would require at least two passes through the stream. In any event, they don't talk about this as a way to get the lower bound to fix the above complexity to be $\log(k)/\epsilon$. The supplement itself is kind of weird, it is mostly the same as the main paper, but in a few cases it is shorter, and a few cases it is longer. For example, the proof of lemma 3 in the supplement has no more detail than the proof in the main paper. Also, Lemma 6 in the supplement is not the same lemma as Lemma 6 in the main paper (in the main paper, Lemma 6 is about the query complexity while in the supplement, Lemma 6 is on the compute time of the Deletion subroutine).

[Author Response · NeurIPS 2020]

First of all, we would like to thank all the reviewers for their comments. Below we answer to the concerns of each reviewer separately.

**Reviewer** 1**:** Thank you for the review and your comment. We will explain the motivation and the background in details and will give a clear description of the dynamic model and the difference between the streaming and the dynamic setting to improve the presentation of the paper. We will also write a complete related work section and the relation of our work to prior work specially a comparison between our work and the results from KZK18 and MKK17.

**Reviewers** 1 **and** 3**: a discussion for the update time and the memory constraint.** Here we briefly mention the difference between the streaming and the dynamic setting. In the streaming setting the main concern is the space complexity. We often compute a sketch of the input that is revealed in a streaming fashion. At the end of the stream we compute a solution/function using the sketch that we maintained in the course of stream. On the other hand, in the dynamic setting the main complexity is the time. The idea is given the input that is revealed in a streaming fashion, one is interested in seeing the solution and the changes in the solution after every insert or delete. The main motivation is for learning highly dynamic and sensitive data (such as time series) that we need to take an action as soon as we see a shift in the function of the underlying data that we observe. Since we need to react to changes in the solution fast, we need to (re)-compute the solution as fast as we can. Indeed, we cannot wait till the end of the stream and take the corresponding action afterwards. The underlying assumption for the dynamic setting is nowadays with machines that can easily have (SD)RAMs of GBs and soon TBs, the space constraint won't be a problem, but the time complexity is the main bottleneck. The results from KZK18 and MKK17 are streaming algorithms whose time complexities depend on the number of deletions (Theorem 1 of the second reference) which will be high if we want to (re)-compute a solution after each insertion or deletion.

**Reviewer** 2**:** Thank you for your comments. We find them useful and will incorporate them to improve the presentation of the paper. Yes, the assumption that the function $f$ is monotone has been used in the related results from KZK18 and MKK17. Also, in Lemma 3 we used the fact that the function must be monotone, but indeed it is a good question to see if we can develop dynamic algorithms for non-monotone functions and we will think about it. As for the method of getting a worst case bound in Section 3, we did not know about "A Deamortization Approach for Dynamic Spanner and Dynamic Maximal Matching". Definitely we will cite this paper.

**Reviewer** 3**:** Thank you for the review and the comments. In the appendix we mentioned a version of our algorithm that doesn't need to know the OPT value and has $O(\sqrt{n})$ time complexity. We recently found that a variant of our algorithm that does not need to know the OPT value and has poly-logarithmic dependency on $n$. The idea is to do our logarithmic rate of sampling and filtering besides maintaining a max-heap as we see new updates. We will add this new algorithm to the paper.

**Reviewer** 4**:** Thank you for your comments. We find them useful and will incorporate them to improve the presentation of the paper. Yes, the running time of our algorithms is measured on the number of oracle calls to the function $f$ and we will explain it explicitly in the paper. We do not know if $1/2$ is the best that we can achieve or not. However, we think it will be very interesting to see if we can develop a dynamic algorithm with better than $1/2$-approximation guarantee. Many thanks for pointing out to the new arXiv submission "Fully Dynamic Algorithm for Constrained Submodular Optimization". We did not know about this paper. We should mention that this paper presents a dynamic algorithm whose expected update time is poly-logarithmic in $n$ and $k$. Our algorithm works with high probability. We think we can use our worst case framework in Section 3 to improve their running time bound from expected to a high probability bound.

**Reviewers** 2 **and** 4**: the dependence on** $k$ **in the running time.** we recently realized that a simple version of our algorithm has poly-logarithmic dependence on $k$ in the running time. The idea is if at each step $i$ of the recursion we sample $O(\epsilon^{-2} k \log n)$ elements and if we have many elements that are above the threshold, we collect more than one such element. We can then show that the number of elements that are survived after filtering at each step $i$ drops by a factor of $\frac{\epsilon^{-2}}{2k}$. Thus, every insert or delete that changes a partial solution set $G_i$ happens with probability $O(n/k)$, but the number of elements in the steps $> i$ are an order of $O(n/k)$ for which we have enough credit to re-run the following steps.



[Meta-Review · NeurIPS 2020]

R3 has a weakly negative opinion on the paper; R1 and R2 have a weakly positive opinion of the paper, and R4 has a positive opinion. In the rebuttal, the authors have responded to some of the points raised by the reviewers; the response has partially satisfied the reviewers --- some points, though, remain. For instance, it appears the assumption of knowing an upper bound on the value of OPT is too strong in many cases; while the authors do consider weaker assumptions, their results for such other assumptions should be presented in more detail. It would be useful to compare the results of this paper with those in https://arxiv.org/abs/2006.04704 , a paper that appeared on the arxiv after the NeurIPS submission deadline --- as per NeurIPS's policy, the decision on this submission was made independently of this "contemporaneous" arxiv paper.